# Fast and efficient Once-For-All Networks for Diverse Hardware Deployment

## Abstract

Convolutional neural networks are widely used in practical application in many diverse environments. Each different environment requires a different optimized network to maximize accuracy under its unique hardware constraints and latency requirements. To find models for this varied array of potential deployment targets, once-for-all (OFA) was introduced as a way to simultaneously co-train many models at once, while keeping the total training cost constant. However, the total training cost is very high, requiring up to 1200 GPU-hours. Compound OFA (compOFA) decreased the training cost of OFA by $2\times$ by coupling model dimensions to reduce the search space of possible models by orders of magnitude, while also simplifying the training procedure.

In this work, we continue the effort to reduce the training cost of OFA methods. While both OFA and compOFA use a pre-trained teacher network, we propose an in-place knowledge distillation procedure to train the super-network simultaneously with the sub-networks. Within this in-place distillation framework, we develop an upper-attentive sample technique that reduces the training cost per epoch while maintaining accuracy. Through experiments on ImageNet, we demonstrate that, we can achieve a $2\times$ - $3\times$ ($1.5\times$ - $1.8\times$) reduction in training time compared to the state of the art OFA and compOFA, respectively, without loss of optimality.

## 1 Introduction

Convolutional neural networks (CNNs) are overwhelmingly successful in many machine learning applications. These applications may have different inference constraints (e.g., latency) and are deployed in different hardware platforms that range from server-grade platforms to edge devices such as smartphones. Optimal network architectures need to be designed to meet the requirement for a target deployment scenario. However, naively designing a specialized architecture for each scenario is very expensive as it requires to fully retrain the model each time. This is an excessively expensive process in terms of the required machine learning expertise, time, energy and $CO_2$ emission.

Recently, researchers have proposed efficient methods which are based on training a super network only once. Then, for a specific deployment scenario, a sub-network is sampled from the super-network that meet the deployment constraints with the best accuracy. The weight of the sampled network is shared with original super network, hence retraining is not required. Once-for-all (OFA) (Cai et al., 2020) is among the first methods proposed to tackle this problem. The OFA method trains a once-for-all network that jointly optimizes the accuracy of a large number of sub-networks (more than $10^{19}$) sampled from the once-for-all network. Each sub-network is selected from the once-for-all network where layer depths, channel widths, kernel sizes and input resolution are scaled independently. Such scaling provides a family of CNNs networks with different computation and representation power to flexibly support deployment under diverse platforms and configurations. With this massive search space, OFA co-trains all the sub-networks by a complex four-stage progressive training process which is prohibitively expensive and costs around 1200 GPU hours.

Compound OFA (CompOFA) (Sahni et al., 2020) builds upon the original OFA by shrinking the design space of possible sub-networks. This is done by only considering networks whose dimensions are coupled. This reduces the number of possible models by 17 orders of magnitudes, from $10^{19}$ down to 243. Sahni et al. (2020) demonstrate that this smaller design space is sufficient, as most

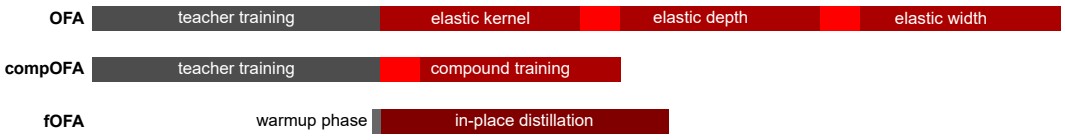

Figure 1: Comparison of training schedules for OFA, CompOFA, and fOFA. Length on the horizontal axis is proportional to the number of epochs in each phase. For OFA, "elastic kernel," "elastic width," and "elastic depth" are the phases of training specified in Cai et al. (2020) that are not used in CompOFA and fOFA.

sub-networks in the original OFA design space are far from the optimal accuracy-latency frontier. With this smaller space, the training procedure can be simplified as well as these suboptimal sub-networks are no longer influencing the training process. CompOFA reduces the four stages of the original OFA process to two stages, and this optimization speeds up the training time of CompOFA by a factor of $2\times$ over OFA.

However, $2\times$ faster than OFA's 1200 GPU-hours is still 600 GPU-hours. Even with this significant improvement, the training cost remains vary expensive, especially when effects on the environment are considered (Strubell et al., 2019). While some of this cost can be mitigated by improvements in hardware efficiency and the continued development of specialized platforms for training CNNs, algorithmic enhancements still have a large role to play. While CompOFA greatly simplifies the progressive shrinking training procedure used in OFA, it is still dependent on pre-training a supernetwork to act as a teacher for the sub-network co-training process, which uses knowledge distillation Hinton et al. (2015). Due to the optimizations in the co-training process, training the supernetwork in CompOFA requires more than half (180 out of 330) of the total training epochs.

In this work, we propose several optimizations to the once-for-all training process that produces a one-stage training algorithm for fast and efficient neural architecture search. The key features of our method are:

- We co-train all of the sub-networks from scratch without pre-training a teacher network, using the concept of in-place distillation (Yu et al., 2020). The largest network we train using in-place distillation is smaller than the pre-trained teacher network used in Cai et al. (2020) and Sahni et al. (2020).

- During the co-training process, we develop an upper-attentive sampling method which always sample the full-sized sub-net at each iteration to help co-train the rest sub-networks.

- Before co-training, we use an upper-attentive warmup technique which trains only the full-sized sub-net for a few epochs before co-training to further improve the performance.

- With these optimizations, we can decrease the number of sampled sub-networks in each iteration of training, further improving performance.

The benefits of our proposed fast OFA (fOFA) method are shown in Figure 1. Furthermore, since our method has only a single stage, we can easily increase the training time and improve on the accuracy of previous methods while still requiring less training time.

The rest of this paper is organized as follows. We describe related work in more detail in Section 2 and illustrate our method in depth in Section 3. We report on our experimental results in Section 4 and finish the paper with conclusions in 6.

## 2 RELATED WORK

Neural architecture search (NAS) aims to automatically find the optimal network architecture given hardware constraints, such as FLOPs or latency. Early NAS works mainly adapted reinforcement learning (Zoph et al., 2018; Zoph & Le, 2016), evolutionary search (Real et al., 2019; 2017), or sparse connection learning (Kim et al., 2018) to sample different architectures. However, each sampled architecture needed to be trained from scratch, resulting in a huge and intractable computing cost. More recent NAS works greatly reduce the cost by training an over-parameterized network

named a *super-network*, and then sample various sub-networks which share the weights with the *super-network*. Such *super-network*-based methods can be further divided into two main categories as follows:

## 2.1 TWO-STAGE TRAINING

The main idea of the two-stage training methods (Berman et al., 2020; Bender et al., 2018; Brock et al., 2017; Guo et al., 2019; Liu et al., 2018; Pham et al., 2018) is that after searching for the best architectures in the first stage of training, the best architectures then have to be retrained from scratch to obtain a final model. Generally, a single two-stage search experiment can only target a single resource budget or a narrow range of resource budgets at a time, which is inefficient.

## 2.2 ONE-STAGE TRAINING

To alleviate the inefficiency of two-stage training, Once-For-All (OFA) (Cai et al., 2020) was proposed to jointly train various sub-networks of the super-network in a single stage. By doing so, the sub-networks could be directly deployed into different hardware platform without retraining. However, to support an extremely large number of sub-networks (i.e., $10^{19}$), such one-stage training involves multi-steps to gradually add more sub-networks by using the proposed progressive shrinking technique. Moreover, OFA also needs to train a single full-sized teacher network (same size as super-network) from scratch firstly to guide the training of the sub-networks by using knowledge distillation. Thus, due to the complex training procedure, OFA still suffers from a prohibitive training cost, requiring around 1200 GPU hours.

More recently, inspired by studies on neural network design spaces (Tan & Le, 2019; Radosavovic et al., 2020), CompOFA (Sahni et al., 2020) proposes a compound sub-network scaling method, which couples the depth and width configuration of the sampled sub-networks to constrain the search space, reducing the space to only 243 number of sub-networks without losing accuracy. Although CompOFA achieves a $2\times$ training cost reduction compared with OFA, it follows the similar training procedure that also needs to train a teacher model from scratch. A similar approach is investigated in Yang et al. (2020), which couples network width and input resolution into a single mutual learning framework.

In addition, bigNAS (Yu et al., 2020) proposed replacing OFA's multi-step training by a single step, namely *one-shot NAS*, challenging the usual practice of progressive training in OFA. The idea of the one-shot NAS is to jointly train sub-networks from scratch directly by using the sandwich rule and in-place distillation techniques proposed for slimmable networks (Yu et al., 2018). Based on bigNAS, Wang et al. (2020) points out the unnecessary updates on sub-optimal models in one-stage training, and uses attention mechanisms to push the Pareto front. However, the primary objective of these two works is to obtain better accuracy, thus still suffer from high training cost, e.g., bigNAS needs over 2300 TPU hours for $O(10^{12})$ models. In addition, (Li et al., 2021) works to improve the trade-off between accuracy and computation complexity based for slimmable networks by introducing a dynamic gating mechanism and in-place ensemble bootstrapping to increase training stability.. However, it requires a one more gating training step, resulting in larger training cost.

Our approach also follows the direction of the *one-shot* model. Differentiating from bigNAS, the primary objective of this work is to reduce the training cost without loss of Pareto optimality under the design space of OFA and CompOFA.

## 3 METHODS

### 3.1 BUILDING THE SEARCH SPACE

A neural network $\mathcal{N}$ is a function that takes an input set $X$ and generates a set of outputs $\delta(\mathcal{N}, X)$. In this work, we focus on a fixed input set (i.e., ImageNet), and thus write the network output as $\delta(\mathcal{N})$. In the supervised learning setting, the performance of the neural network is evaluated against a set of labels $Y_D$.

Following the standard practice in neural architecture search, we limit our neural network space to the set of architectures that consists of a sequence of blocks $B_1, B_2, \ldots, B_m$, where $m = 5$ is a

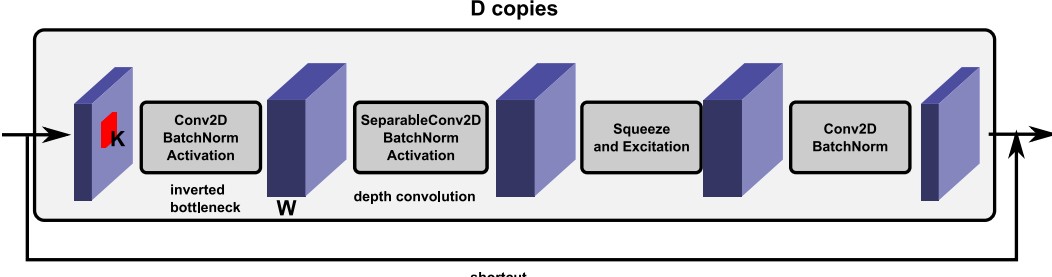

Figure 2: The search space used for fOFA, based on the architecture space of MobileNetV3 (Howard et al., 2019). The dimension $K$ refers to the size of the convolutional kernel, $W$ to the channel expansion ratio, and $D$ to the number of repetitions of the block.

typical value. Each block is based on the inverted residual in the architecture space of MobileNetV3 (Howard et al., 2019). A block is parameterized by three dimensions: the depth (number of layers in the block) $D$, the width (channel expansion ratio) $W$, and the convolution kernel size $K$. This search space is illustrated in Figure 2.

To reduce the size of the search space, we use the same coupling heuristic as CompOFA (Sahni et al., 2020); that is, if there are $n$ choices for the depth dimension and $n$ choices for the width dimension, we sample the $i$th largest depth $w_i$ whenever we sample the $i$th largest depth $d_i$ for each layer in the block. While OFA uses an *elastic kernel* that allows for different kernel sizes within blocks, we follow CompOFA and use a *fixed kernel* size within each block. We call the network where the values of $K$, $D$, and $W$ are each their largest possible value the *full-sized network* or *super-net*, and the network created by any other choice of these values a *sub-network*.

As in CompOFA, we choose three possible values for $D \in \{2, 3, 4\}$ and three possible values for $W \in \{3, 4, 6\}$ and fix the kernel size to that of (Howard et al., 2019), that is, $K = 3$ in the first, third, and fourth blocks, and $K = 5$ in the second and fifth blocks. Thus, with five blocks, we have $3^5 = 243$ models in our search space.

In neural architecture search, the input resolution can vary as well, up to a maximum size of $224 \times 224$ for ImageNet (Deng et al., 2009). In this work, we use an elastic resolution, where input images are resized to be square with dimension in the set $\{128, 160, 192, 224\}$.

## 3.2 ONCE-FOR-ALL TRAINING

Both OFA and CompOFA use knowledge distillation to guide the super-net co-training procedure. In general, co-training all the sub-networks with a teacher model can be considered as a multi-objective optimization problem, which can be formulated as:

$$\min_{\mathcal{N}} \sum_{a_i} \mathcal{L}\left(\mathcal{N}_{a_i}, \mathcal{N}_T, Y_D\right), \tag{1}$$

where $\mathcal{N}$ denotes the weights of the full-sized network, $\mathcal{N}_T$ is the additional pre-trained teacher model, and $\mathcal{N}_{a_i}$ is a random sub-network of $\mathcal{N}$ where $a_i$ specified the sub-network architecture. The loss function $\mathcal{L}$ is

$$\mathcal{L}\left(\mathcal{N}_{a_i}, \mathcal{N}_T, Y_D\right) = \mathcal{L}\left(\delta\left(\mathcal{N}_{a_i}\right), Y_D\right) + \beta * \mathcal{L}\left(\delta\left(\mathcal{N}_{a_i}\right) + \delta\left(\mathcal{N}_T\right)\right) \tag{2}$$

where $\beta$ denotes the distillation weight. This optimization function aims to co-train all the sub-networks during the training using both the target label and output of the teacher network using knowledge distillation. However, because there are so many sub-networks, it is not practical to compute this loss function in its entirety. So, following the approach of OFA and CompOFA, we randomly sample $n$ sub-networks in each training iteration. The loss function is thus reformulated as

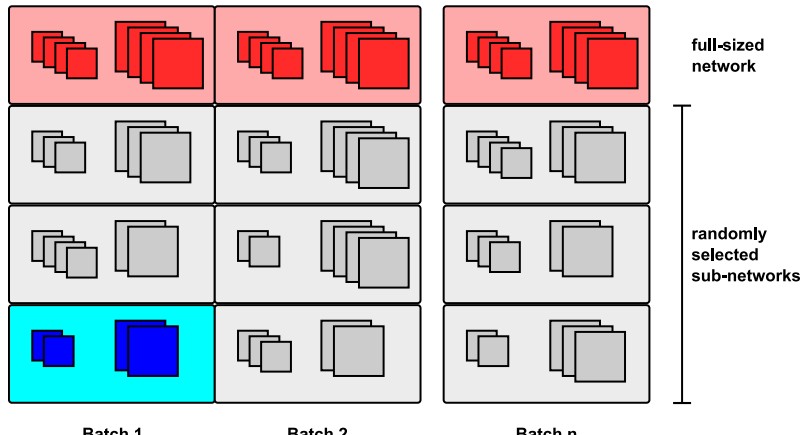

Figure 3: In upper attentive sampling, the largest possible model (in our case with $D = 4$ and $W = 6$ for all blocks, shown in red) is selected during each batch of the training process. The other models selected at each batch are randomly chosen from all possible sub-networks. Upper attentive sampling differs from the "sandwich model" of Yu et al. (2020) in that the smallest possible model (shown in blue) need not be selected at each batch.

$$\min_{\mathcal{N}} \sum_{i}^{n} \mathcal{L}\left(\mathcal{N}_{rand(a_i)}, W_T, Y_D\right), \tag{3}$$

where $n = 4$ is a typical value of the number of sub-networks to sample.

### 3.3 IN-PLACE DISTILLATION

Requiring the training of a teacher model $\mathcal{N}_T$ adds significant overhead to the total training time, as teacher training must be completed before the training of subnets can begin. (Sahni et al., 2020) reports that teacher training takes up 17.6% of the wall time for OFA and 35.0% for CompOFA.

In this work, we propose to eliminate training the teacher model and instead co-training the sub-networks from scratch. If we remove $\mathcal{N}_T$ from the loss function above, we reformulate a random sampling loss function

$$\min_{\mathcal{N}} \sum_{i}^{n} \mathcal{L}\left(\mathcal{N}_{rand(a_i)}, Y_D\right), \tag{4}$$

where $\mathcal{L}\left(\mathcal{N}_{a_i}, Y_D\right) = \mathcal{L}\left(\delta\left(\mathcal{N}_{a_i}\right), Y_D\right)$ for any network $\mathcal{N}_{a_i}$. However, this naive sampling method results in significant accuracy drops if co-training sub-networks from scratch. To improve accuracy, BigNAS (Yu et al., 2020) uses the "sandwich model" from (Yu & Huang, 2019), wherein the largest and smallest possible sub-networks are always sampled. Its loss function is

$$\min_{\mathcal{N}} \left( \mathcal{L}\left(\mathcal{N}_{\max}, Y_D\right) + \sum_{i=1}^{n-2} \mathcal{L}\left(\mathcal{N}_{rand(a_i)}, \mathcal{N}_{\max}\right) + \mathcal{L}\left(\mathcal{N}_{\min}, \mathcal{N}_{\max}\right) \right), \tag{5}$$

where $\mathcal{N}_{\max}$ denotes the full-sized network and $\mathcal{N}_{\min}$ denotes the smallest sub-network. The full-sized network is thus trained in parallel with the smaller models.

### 3.4 UPPER ATTENTIVE SAMPLING

The sandwich model proposed for BigNAS applies to a high training cost scenario where $10^{12}$ models are being evaluated. In our scenario, with only 243 models, we find that including the

Table 1: Training schedule for fOFA. We replace the lengthy teacher training phase with in-place distillation, preceded by a short warmup phase. We also decrease the size of the teacher kernels from $K = 7$ to $K = 3$ or $K = 5$, as described in Section 3.1.

| Method | Phase | K | D | W | $N_{sample}$ | Epochs |
|---|---|---|---|---|---|---|
| | Teacher | 7 | 4 | 6 | 1 | 180 |
| CompOFA | Compound | 3/5 | 2, 3, 4 | 3, 4, 6 | 4 | 25 |
| | Compound | 3/5 | 2, 3, 4 | 3, 4, 6 | 4 | 125 |
| | Warmup | 3/5 | 4 | 6 | 1 | 5 |
| fOFA | In-place Distillation | 3/5 | 4 | 6 | 1 | 180 |
| | (simultaneous) | 3/5 | 2, 3, 4 | 3, 4, 6 | 2, 3 | 180 |

smallest model $\mathcal{N}_{\min}$ adversely affects the overall accuracy. To address this issue, we develop a new upper-attentive sampling method, which always samples the full-sized sub-network in each iteration, and $n - 1$ random sub-networks. The loss function of upper-attentive sampling is:

$$\min_{\mathcal{N}} \left( \mathcal{L}\left(\mathcal{N}_{\max}, Y_D\right) + \sum_{i=1}^{n-1} \mathcal{L}\left(\mathcal{N}_{rand(a_i)}, \mathcal{N}_{\max}\right) \right) \qquad (6)$$

Where $\mathcal{N}_{\max}$ represents the largest sub-network. During training, the largest sub-network is maximized only with respect to the ground truth labels, while the additional sub-networks are trained with respect to the output of the largest sub-network.

A schematic of upper attentive sampling is shown in Figure 3. With upper attentive sampling, we may either choose to replace the smallest model with a randomly selected sub-network, or choose to remove it entirely, effectively reducing the number of sampled networks by 1 when compared to CompOFA or BigNAS. Intuitively, removing the smallest network will result in faster training than replacing it, but may have a negative impact on accuracy. We study both the removing and replacing options in Section 4.

### 3.5   WARMUP PHASE

Because the full-sized sub-network $\mathcal{N}_{\max}$ is a soft target for the other sub-networks, we find training benefits from a warmup phase so that the initial target for the smaller sub-networks is not random at the start. We find that first training the largest sub-network for a few epochs provides good results and is still much faster than training a teacher from scratch for 180 epochs.

### 3.6   SUB-NETWORK SELECTION PROCEDURE

Again, following (Sahni et al., 2020) and (Cai et al., 2020), we use evolutionary search (Real et al., 2019) to retrieve specific sub-networks that are optimized for a given hardware target. This search finds trained networks that maximize accuracy subject to the target latency or FLOP constraint. For hardware targets such as the Samsung Note10, latency can be estimated using a look-up table estimator from (Cai et al., 2020).

## 4   RESULTS

### 4.1   TRAINING SETUP

We performed our experiments on an NVIDIA DGX-A100 server with 8 GPUs. Experiments were run in version 21.03 of the NVIDIA NGC pytorch container[1], which includes Python 3.8, pytorch 1.9.0, and NVIDIA CUDA 11.2. Horovod version 0.19.3 was used for multi-GPU training.

---

[1]`https://docs.nvidia.com/deeplearning/frameworks/pytorch-release-notes/rel_21-03.html`

The training schedule for fOFA is listed in Table 1 and comparisons with CompOFA and fOFA are shown in Figure 1. CompOFA requires 330 total epochs, of which over half (180) are dedicated to training the full-size teacher model. For fOFA, we require 185 total epochs. Of these, only 5 epochs are used for warming up the full-size model in advance of in-place distillation, wherein the supernet and the randomly selected networks are trained simultaneously.

Directly following CompOFA (Sahni et al., 2020) , we perform the model search over the MobileNetV3 space with expansion ratio $1.0^2$. We use 8 GPUs, a batch size of 256 per GPU, and a learning rate of 0.325. For a fair comparison, all other hyper-parameters are set to the same values as OFA and CompOFA, including a cosine learning rate schedule, momentum of 0.9, batch-norm momentum of 0.1, weight decay of 3e-5, label smoothing of 0.1, dropout rate of 0.1. Also, as fOFA is trained from scratch instead of fine-tuning on the pre-trained teacher model, a gradient clipping threshold of 1.0 is adapted to make the training stable, following bigNAS (Yu et al., 2020).

## 4.2 TRAINING RESULTS

Table 2: Mean Top-1 Accuracy on ImageNet

| Method | Epochs | Training Cost (GPU h) | Mean Top-1 Accuracy |
|---|---|---|---|
| OFA | 605 | 672 (1.0x) | 75.5 |
| CompOFA | 330 | 336 (2.0x) | 75.4 |
| fOFA (n=3) | **185** | **184 (3.7x)** | 75.5 |
| fOFA (n=4) | **185** | 216 (3.1x) | 75.5 |
| fOFA (n=4) | 300 | 350 (1.9x) | **75.6** |

Table 2 shows the average accuracy over the generated models. For fOFA, $n = 3$ means that the smallest model from the sandwich rule of (Yu et al., 2020) has been removed, and we are training with the largest model and two randomly selected sub-networks. $n = 4$ means that the smallest model has been replaced, and we are training with the largest model and three randomly selected sub-networks.

For CompOFA and fOFA, the mean top-1 accuracy is computed over all 243 models generated by the training process. Since OFA has an extremely large number of sub-networks, we calculate this average by selecting the same 243 models that are used in CompOFA and fOFA. We see that while CompOFA is $2.0\times$ faster than OFA, with 185 epochs, fOFA is a further $1.55\times$ faster than CompOFA if we sample four sub-networks during training, and a further $1.83\times$ faster if we sample only three sub-networks. In both cases, our accuracy is equal to OFA and 0.1% greater than CompOFA. If we sample four sub-networks and extend our training time to approximately match the number of GPU-hours required for CompOFA, we generate an average accuracy 0.1% greater than OFA.

## 4.3 HARDWARE LATENCY

Table 3: Top-1 Accuracy on Latency Constrained Models for a Samsung Note10

| Method | Epochs | Latency Constraint | | | |
|---|---|---|---|---|---|
| | | 15 ms | 20 ms | 25 ms | 30 ms |
| OFA | 605 | 71.93 | 73.95 | 74.94 | 75.41 |
| compOFA | 330 | 72.08 | 73.94 | 74.94 | 75.58 |
| fOFA (n=3) | **185** | 72.02 | 74.06 | 75.06 | 75.60 |
| fOFA (n=4) | **185** | 71.74 | 73.78 | 74.77 | 75.47 |

Table 3 shows the performance of the once-for-all methods for the hardware deployment scenario of a Samsung Note10. We used the latency estimator for the Note10 CPU provided by Cai et al. (2020). The latency thresholds of 15, 20, 25, and 30 milliseconds are selected to match the latency targets

---

[2]In Cai et al. (2020), results are reported with an expansion ratio of 1.2, so the numbers reported here are not identical.

used in Sahni et al. (2020). At latencies larger than 20 ms, fOFA with $n = 3$ is more accurate than other methods while also having the smallest training cost. At 15 ms, fOFA is slightly less accurate than compOFA, but is still $1.83\times$ faster in training time.

Table 4: Top-1 Accuracy on Latency Constrained Models for GPU platforms.

| Method | Epochs | A100 Latency Constraint | | | | Pascal Latency Constraint | | | |
|---|---|---|---|---|---|---|---|---|---|
| | | 4 ms | 6 ms | 8 ms | 10 ms | 15 ms | 25 ms | 35 ms | 40 ms |
| OFA | 605 | 72.55 | 76.12 | 77.02 | 77.11 | 73.86 | 76.11 | 77.00 | 77.10 |
| compOFA | 330 | 72.78 | 76.26 | 77.40 | 77.49 | 72.84 | 76.27 | 77.45 | 77.52 |
| fOFA (n=3) | **185** | 73.63 | 76.08 | 77.39 | 77.55 | 74.46 | 76.28 | 77.37 | 77.46 |
| fOFA (n=4) | **185** | 73.75 | 76.33 | 77.21 | 77.28 | 74.51 | 76.29 | 77.15 | 77.36 |

Table 4 (left) shows the performance of the methods on a NVIDIA A100 GPU. In this setting, the latency is measured directly using the CompOFA code[3]. fOFA with $n = 4$ has the highest accuracy at the strictest latency constraints (4 ms and 6 ms), while fOFA with $n = 3$ performs best at 10 ms latency. fOFA (n=3) and compOFA have nearly identical accuracy at 8 ms. In Table 4 (right), we also show the performance of the methods for an earlier-generation GPU (Nvidia Pascal) with more relaxed latency constrains. The results show similar trends to the A100 GPU where fOFA is superior at the strictest constraints and has similar accuracy to CompOFA at high constraints.

Table 5: Top-1 Accuracy on Latency Constrained Models for an AMD EPYC 7763 CPU

| Method | Epochs | Latency Constraint | | | |
|---|---|---|---|---|---|
| | | 22 ms | 25 ms | 28 ms | 31 ms |
| OFA | 605 | 74.34 | 74.92 | 74.92 | 76.05 |
| compOFA | 330 | 72.77 | 74.65 | 75.17 | 76.35 |
| fOFA (n=3) | **185** | 73.55 | 75.01 | 75.38 | 75.78 |
| fOFA (n=4) | **185** | 73.44 | 74.69 | 75.59 | 75.85 |

Table 5 shows the performance of the methods on CPU. Again, latency is measure directly using the CompOFA code. In this setting, we find that fOFA achieves the highest accuracy at medium constraints (25 ms and 28 ms), while compOFA achieves the best accuracy at 31ms, and OFA as 22 ms.

## 5   DISCUSSION

Figure 4 shows the trade-off between model accuracy and number of floating-point operations for compOFA and fOFA with $n = 4$. On average, fOFA is $\sim 0.1\%$ more accurate that CompOFA, as listed in Table 2, and achieves greater accuracy on models with lower FLOP counts, agreeing with results in Tables 3-5. Despite replacing the sandwich rule with upper-attentive sampling, the smallest model in the search space has 0.9% greater accuracy in fOFA.

We also ran fOFA using the sandwich rule (Yu & Huang (2019)) on the same hyperparameter space. With the sandwich rule, the average accuracy over the search space was 0.3% lower than with upper-attentive sampling. Furthermore, the decrease in accuracy was greater on models with higher FLOP counts, and less on models with lower FLOP counts. To explain these observations, we note that the upper bound of the CompOFA search space is significant lower than that of the MobileNetv2 search space from Yu et al. (2020). In BigNAS, the largest model in the search space required 1.8 GFLOPs while the largest output model, BigNASModel-XL, required only 1.04 GFLOPs. In contrast, the largest model in the CompOFA search space uses 447 MFLOPs and the models we selected for GPU deployment in Table 4 approach this upper limit.

In our experiments, we set the convolution kernel sizes for each block to those used in MobileNetV3 (Howard et al. (2019)) for all models in the search space, including the teacher. We also experimented increasing the size of the teacher model, using $K = 7$ for each block in the the teacher, and

---

[3]https://github.com/gatech-sysml/CompOFA/tree/main/ofa

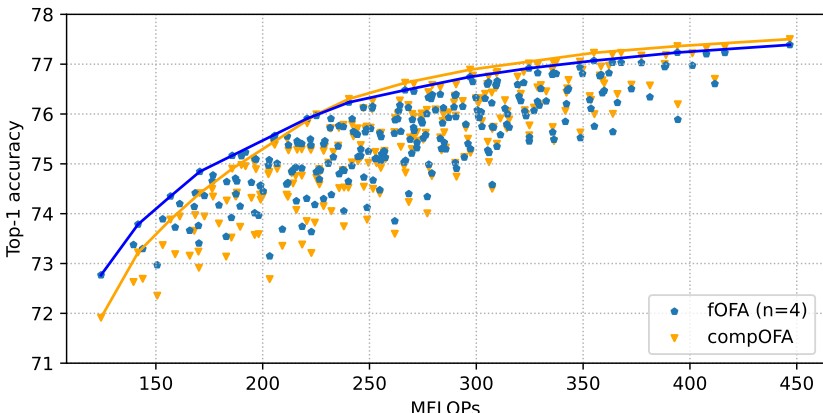

Figure 4: Comparison of floating point operation vs. accuracy for each of the 243 models in the compOFA searchspace, for compOFA (orange) and fOFA with $n = 4$ (blue).

found that this results in the average accuracy decreasing to 74.8%. From this result, we propose that an overly large teacher model, while providing a higher upper bound on accuracy, may not be as effective for training smaller submodels, and that when the teacher model is closer in size to the submodels, upper-attentive sampling is sufficient to achieve good accuracy throughout the search space.

When upper-attentive sampling is used in combination with in-place distillation, the warm-up phase is essential so that the initial target for sub-model training is better than random. After five epochs of warm-up, the teacher model has an accuracy of 47.42% in our experiments, providing a reasonable starting point for training. In OFA and CompOFA, this warmup phase is not needed because the teacher model is already fully trained.

## 6 CONCLUSION

In this work, we introduce fast once-for-all (fOFA) and demonstrate how this methodology can reduce expensive training cost for neural architecture search below 200 GPU-hours. This is done by combining the approaches of CompOFA (Sahni et al., 2020) and BigNAS (Yu et al., 2020) by using a reduced state space to only consider models close to the Pareto-optimum, as in CompOFA, and using in-place distillation, as in BigNAS, to eliminate the requirement of expensive teacher training from OFA. While these methods work well together, to achieve optimal performance it was necessary to develop methods such as upper attentive sampling and apply a warmup phase to achieve optimal results. Our results show that we can achieve the same accuracy as OFA with a speed-up of 3.1×-3.7×, and similar accuracy to CompOFA with a speed-up of 1.5×-1.8×.

As we continue this work, we look to further investigate the relationships between training heuristics and the choice of the network search space, so that we can better understand the theoretical reasons for their performance and determine which methods lead to most efficient training. We also hope to further investigate multiple network search spaces to develop new methods for finding training schedules and network sampling approaches that optimize accuracy for both benchmark tasks such as Imagenet classification and novel tasks such as segmentation, detection, and transfer learning.

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
