# OpenReview forum: "Fast and Efficient Once-For-All Networks for Diverse Hardware Deployment"
_ICLR.cc/2022/Conference — ICLR 2022 Submitted_

### Official Review · Reviewer_vCfG · 2021-10-25

**Correctness:** 3
**Technical Novelty And Significance:** 2
**Empirical Novelty And Significance:** 2
**Recommendation:** 3
**Confidence:** 4

**Main Review:**

Strength:
1. The proposed method improved the training speed without loss of accuracy.
2. The paper is easy to follow.

Weakness:
1. The proposed method is of very limited novelty and technical contribution. Looking at the contribution part, (1) in-place distillation is used in Slimmable Network. (2) The upper-attentive sampling is basically Sand-wich rule in Slimmable Network, the only difference is that the author doesn’t sample the smallest sub-network. (3) The warm-up training is also a very incremental and commonly-used technique. The proposed method is basically a combination of previous techniques.
2. The results seem to be not consistent with that reported in the paper. For example, in Table 3, the performance of OFA seems to be lower than that reported in the original paper.
3. Some missing related works: \
[1] NestedNet: Learning Nested Sparse Structures in Deep Neural Networks \
[2] MutualNet: Adaptive ConvNet via Mutual Learning from Network Width and Resolution \
[3] Dynamic Slimmable Network


**Summary Of The Paper:**

This paper proposed a method to train a once-for-all network, where one network can run at different resource constraints. The method is based on previous methods, and the author further improved the training speed by around 1.5x - 1.8x without loss of performance. The method is evaluated on ImageNet classification.

**Summary Of The Review:**

Overall, I think the paper has limited technical contribution, and the training speedup is not surprising (less than 2x). If it is a combination of techniques but significantly improved the training speed (say 5x-10x), which makes such network training easily-doable for everyone, then I would say it is a large contribution. Currently, I think the author need to further improve their method.

======== Post Rebuttal ========

Thank the author for the response. My concern about the novelty and significance of the method and results still remain. I agree that combining existing techniques may need more tuning. But this is hard to be considered as enough contribution, especially given that the performance is not significantly improved. Therefore, I will keep my score.

---

> ### Author Response · Authors · 2021-11-22
> **Response to Reviewer vCfG**
>
> In addition to the comments listed in the overall response, we would like to directly address the following points:
> 1. “The proposed method is of very limited novelty and technical contribution. Looking at the contribution part, (1) in-place distillation is used in Slimmable Network. (2) The upper-attentive sampling is basically Sand-wich rule in Slimmable Network, the only difference is that the author doesn’t sample the smallest sub-network. (3) The warm-up training is also a very incremental and commonly-used technique. The proposed method is basically a combination of previous techniques.”
>    - Our method is indeed a combination of previous techniques; the objective of our work is to determine which previously applied techniques are most valuable for decreasing the training time of OFA. While this combination of techniques produces slightly less than 2x speed-up in training over compOFA, this is still a significant absolute improvement for OFA techniques, as it results in a savings of 145 GPU hours. From a practical perspective, we are able to drive the training time below 24 hours on a 8-GPU setup, which increases the range of people who can use once-for-all methodology in practice.
> 2. “The results seem to be not consistent with that reported in the paper. For example, in Table 3, the performance of OFA seems to be lower than that reported in the original paper.”
>    - In our approach, to allow for an apples-to-apples comparison, we follow the procedure of CompOFA as much as possible. This means that we use the same design space (MobileNet V3 1.0x) and hyperparameters as given in OFA. You will see that the performance we report for OFA and compOFA agrees with that reported in Figure 3 of the CompOFA paper from Sahni et al. A potential difference between the original OFA paper and the later works is that OFA uses the MobileNetVv3 1.2x space, which would provide for different results. We have amended Section 4.1 of the paper to clarify these points.
> 3. “Some missing related works:”
>    - Thank you for these valuable references. We have incorporated them into the related work section of the revised paper.

---

### Official Review · Reviewer_cquh · 2021-10-30

**Correctness:** 3
**Technical Novelty And Significance:** 2
**Empirical Novelty And Significance:** 2
**Recommendation:** 5
**Confidence:** 4

**Details Of Ethics Concerns:**

Not applied.

**Main Review:**

In this manuscript, the authors propose a new framework for training the once-for-all (OFA) networks. This framework uses an in-place distillation schema to replace the teacher training in the former OFA training fashion and makes the overall training time faster. The paper is overall well-written with good performance.

My concerns are as follows.

- The novelty for this paper is limited: this paper claims four contributions they made, but the idea for training without the teacher network borrowed from the in-place distillation paper without much modification. The upper-attentive schema simply samples the biggest network in each phase, which can be regarded as "merging“ the training for "teacher network" and "elastic network" in one phase, but the author does not show the performance boost for the schema (compared to without using upper-attentive).
- The OFA aims to alleviate search cost across different hardware platforms, but the comparison in the experiment only include little platform. The author should prepare more experiments across different platforms (mobile, CPU, GPU, FPGA, etc...).
- It is not clear how each component in this paper contributes to the final performance for the reduction of training time, the author should do an ablation study to justify this.

I will raise my vote if the concerns are addressed.

**Summary Of The Paper:**

In this manuscript, the authors propose a new framework for training the once-for-all (OFA) networks. This framework uses an in-place distillation schema to replace the teacher training in the former OFA training fashion and makes the overall training time faster. The paper is overall well-written with good performance.

**Summary Of The Review:**

The paper is overall well-written, but the novelty for this paper is kind of limited. The author should prepare more experiments to justify the performance and address the technical contribution for this paper.

I will raise my vote if the concerns are addressed.

---

> ### Author Response · Authors · 2021-11-22
> **Response to Reviewer cquh**
>
> In addition to the comments listed in the overall response, we would like to directly address the following points:
> 1. “The novelty for this paper is limited: this paper claims four contributions they made, but the idea for training without the teacher network borrowed from the in-place distillation paper without much modification. The upper-attentive schema simply samples the biggest network in each phase, which can be regarded as "merging“ the training for "teacher network" and "elastic network" in one phase, but the author does not show the performance boost for the schema (compared to without using upper-attentive).”
>    - We found that using upper-attentive sampling resulted in 0.3% increase in accuracy over using the standard sandwich rule. This heuristic was based on the observation that, with the sandwich rule in place, the fOFA method outperformed CompOFA on the smallest subnets, but underperformed on the larger subnets. We have amended the manuscript to better explain the motivation for upper-attentive sampling in the new Section 5.
> 2. “The OFA aims to alleviate search cost across different hardware platforms, but the comparison in the experiment only include little platform. The author should prepare more experiments across different platforms (mobile, CPU, GPU, FPGA, etc...).”
>    - In our original manuscript, we reported results for one mobile device (the Samsung Note10), one CPU (the AMD EPYC 7763) and one GPU (the NVIDIA A100). During the rebuttal period, we collected data for an additional GPU (NVIDIA P100). We wished to collect results on platforms specified in Cai et al. during the rebuttal time, but found that their published lookup-tables, which would allow us to do this quickly, were no longer available.
> 3. It is not clear how each component in this paper contributes to the final performance for the reduction of training time, the author should do an ablation study to justify this.
>    - We have re-analyzed the experiments completed before initial submission of the paper and these results have allowed us to investigate the effect of certain algorithm choices on performance. We now report on the difference between upper-attentive sampling and the sandwich rule and effects of varying the teacher size in Section 5.

---

### Official Review · Reviewer_vTmv · 2021-11-02

**Correctness:** 3
**Technical Novelty And Significance:** 2
**Empirical Novelty And Significance:** 3
**Recommendation:** 5
**Confidence:** 4

**Main Review:**

Strengths
1. The training cost saving is clear and significant while maintaining the same level of accuracy as OFA and CompOFA.
2. The proposed method can potentially benefit real-world deep learning applications.

Weaknesses:
[major]
1. Limited novelty. Although this paper presents good results, the novelty of the proposed method is a bit weak. For example, in-place distillation is not new. Upper-attentive sampling and upper-attentive warmup are more like engineering tricks.
2. This paper only presents results on one design space (mobilenetv3). But, in practice, MobileNetV3 is not supported or not efficient in many cases (e.g., on FPGA, GPU, etc). Showing the generalizability of the proposed method on other design spaces (e.g., MobileNetV2) is important.

[minor]
1. On ImageNet, it seems that only supporting 243 sub-networks is sufficient for NAS. But it is unclear whether this strategy also works under the transfer learning setting.


**Summary Of The Paper:**

This manuscript aims to reduce the training cost of once-for-all networks. The proposed method is built upon CompOFA that reduces the number of sub-networks within the OFA network to 243 by only considering networks whose dimensions are coupled. This paper introduces several techniques to further reduce the training cost, including in-place distillation, upper-attentive sampling, and upper-attentive warmup.

**Summary Of The Review:**

This paper presents several simple methods to reduce the training cost of once-for-all networks, showing clear and significant training cost saving without losing accuracy. But, I find the novelty of the proposed method is a bit weak, and the generalizability of the method on other design spaces and transfer learning is unclear.

---

> ### Author Response · Authors · 2021-11-22
> **Response to Reviewer vTmv**
>
> In addition to the comments listed in the overall response, we would like to directly address the following points:
> 1. “This paper only presents results on one design space (mobilenetv3). But, in practice, MobileNetV3 is not supported or not efficient in many cases (e.g., on FPGA, GPU, etc). Showing the generalizability of the proposed method on other design spaces (e.g., MobileNetV2) is important.”
>    - We agree that extending this work to additional design spaces is important. Within the scope of this paper, we wish to demonstrate that the paradigm developed by compOFA can be improved upon to accelerate training time for OFA tasks. Unfortunately, the rebuttal period is not long enough to complete lengthy experiments on a new design space. Thus, we have added a discussion section (Section 5)that elaborates on the characteristics of the MobileNetV3/compOFA design space and how it differs from the MobileNetV2 design space examined in BigNAS (Yu et al., 2020).
> 2. “On ImageNet, it seems that only supporting 243 sub-networks is sufficient for NAS. But it is unclear whether this strategy also works under the transfer learning setting.”
>    - ImageNet is the standard data set used in the development for OFA techniques, and we follow the lead of Cai et al. and Sahni et al. in focusing on this particular dataset. Extending this work to settings such as transfer learning is definitely a direction that we should investigate in future work, and we have indicated as such in the revised conclusion.

---

### Official Review · Reviewer_RKoZ · 2021-11-07

**Correctness:** 4
**Technical Novelty And Significance:** 2
**Empirical Novelty And Significance:** 2
**Recommendation:** 6
**Confidence:** 2

**Main Review:**

Strength:
- Overall this paper is well written and easy to follow.
- The reduction in training time is indeed decent

Weakness:
- Limited novelty.
- Lacking the deeper explanation of the improved results.


**Summary Of The Paper:**

This work proposes new methods of Once-for-all(OFA) network training methods, which significantly reduce the training time as compared to the existing OFA methods. Specifically, this work:
- Eliminate the need of pretraining the teacher network with in-place distillation
- Develop an upper-attentive sampling method which always sample the full-sized sub-net at each iteration to help co-train the rest sub-networks
- Use an upper-attentive warmup technique which trains only the full sized sub-net for a few epochs before co-training to further improve the performance

With these optimizations, this work can decrease the number of sampled sub-networks in each iteration of training, further reducing the total training cost.


**Summary Of The Review:**

- **Limited novelty**.
It seems to me the two major contributions in-place distillation and upper-attentive sampling are already proposed by the existing works. It is a little unclear what further modifications the authors did other than stitching these two techniques together. Would be great if the authors can give some stress here considering the improvements in the results.

- **Lacking the deeper explanation of the improved results.**
In the results part, it seems to me that the proposed methods just magically improve as compared to on the baselines. Not clear the underlying reasons and insights for them. Would be nice for the authors to point out as I might miss them.

---

> ### Author Response · Authors · 2021-11-22
> **Response to Reviewer RKoZ**
>
> In addition to the comments listed in the overall response, we would like to directly address the following points:
> 1. “Limited novelty. It seems to me the two major contributions in-place distillation and upper-attentive sampling are already proposed by the existing works. It is a little unclear what further modifications the authors did other than stitching these two techniques together. Would be great if the authors can give some stress here considering the improvements in the results.”
>    - We have added emphasis to explain why these techniques are effective and reducing training time in the design space proposed for CompOFA by Sahni et al. We have added a discussion of how this contrasts with the MobileNetV2 design space used for BigNas by Yu et al. which goes into further detail about how the modifications of the design space are an important factor in interpreting the results. This can be found in Section 5 in the new manuscript.
> 2. “Lacking the deeper explanation of the improved results. In the results part, it seems to me that the proposed methods just magically improve as compared to on the baselines. Not clear the underlying reasons and insights for them. Would be nice for the authors to point out as I might miss them.”
>    - To address these issues, we have added a new discussion section that explains our reasoning as to why these methods work well together and what features of the design space lead to the improved results. This is also in Section 5 in the new manuscript.

---

### Author Response · Authors · 2021-11-22
**Response to all Reviewers**

First and foremost, we thank the reviewers for their time spent on our work and are appreciative of their efforts. While we provide individual responses to all reviews that address specific concerns, there are some points common to multiple reviews that we wish to address collectively.
- The reviewers consistently asked for deeper explanations about the results and why the set of techniques that we propose in this paper are effective at reducing training time while retaining accuracy. We have added a significant discussion section (Section 5) to better address the novelty of our work and the importance of various features of our method. In contrast to many papers in neural architecture search that focus on maximizing accuracy, our work is primarily focused on *reducing training time while maintaining accuracy*. While several specific techniques we use have been applied elsewhere previously, the novelty of our work is demonstrating how they can be used to accelerate training.
- Before submitting our original manuscript, we had performed experiments wherein we used the sandwich rule from Yu et al. instead of the upper-attentive sampling we propose in this work. We found that, in the compOFA search space, upper-attentive sampling outperforms the sandwich rule. The results of this experiment and discussion of these results are now reported in Section 5.
- Again before submitting our original manuscript, we had performed an experiment where we increased the size of the teacher model and found that this results in decreased accuracy using the same learning schedule. The results of this experiment and discussion of these results are now reported in Section 5.
- Due to the limited rebuttal time, we did not run any new experiments after submission that required training new models. At the request of reviewer cquh, we added an additional hardware target to the results in Section 4, as this could be done using the outputs of the previously completed training process.

---

### Decision · Program_Chairs · 2022-01-20

**Decision:**

Reject

**Comment:**

This paper exposes a method to reduce the training cost of once-for-all networks.
Overall this paper is well written and easy to follow, and the experimental section shows a clear reduction of training time on the examples used.
However, the reviewers point out that the experimental section could benefit from adding more design spaces, and have a better explanation of the results. More importantly, three out of four reviewers agree that the novelty of this work is too low for the submission to be accepted, with the fourth one only giving a score of 6 (and also noting the lack of novelty). I therefore recommend reject for this paper.